# Methane Point Source Quantification Using MethaneAIR: A New Airborne Imaging Spectrometer

Apisada Chulakadabba[1], Maryann Sargent[1], Thomas Lauvaux[2], Joshua S. Benmergui[1, 3, 4], Jonathan E. Franklin[1], Christopher Chan Miller[1, 5], Jonas S. Wilzewski[1, 5], Sébastien Roche[1, 5], Eamon Conway[5, 6], Amir H. Souri[5], Kang Sun[7, 8], Bingkun Luo[5], Jacob Hawthrone[5], Jenna Samra[5], Bruce C. Daube[1], Xiong Liu[5], Kelly Chance[5], Yang Li[9], Ritesh Gautam[3, 4], Mark Omara[3, 4], Jeff S. Rutherford[10], Evan D. Sherwin[10], Adam Brandt[10], and Steven C. Wofsy[1]

[1]Harvard John A. Paulson School of Engineering and Applied Sciences, Harvard University, Cambridge, MA, USA
[2]Molecular and Atmospheric Spectrometry Group (GSMA) – UMR 7331, University of Reims Champagne Ardenne, France
[3]Environmental Defense Fund, New York, NY
[4]MethaneSAT, LLC, Austin, TX
[5]Center for Astrophysics | Harvard & Smithsonian, Cambridge, MA
[6]Kostas Research Institute for Homeland Security, Northeastern University, Burlington, MA, USA
[7]Department of Civil, Structural and Environmental Engineering, University at Buffalo, Buffalo, NY, USA
[8]Research and Education in Energy, Environment and Water Institute, University at Buffalo, Buffalo, NY, USA
[9]Department of Environmental Science, Baylor University, Waco, TX
[10]Department of Energy Science & Engineering, Stanford University, Stanford, CA

**Correspondence:** Apisada Chulakadabba (achulakadabba@seas.harvard.edu)

**Abstract.** The MethaneSAT satellite instrument and its aircraft precursor, MethaneAIR, are imaging spectrometers designed to measure methane concentrations with wide spatial coverage, fine spatial resolution, and high precision compared to currently deployed remote sensing instruments. At 12960 m cruise altitude above ground (13850 above sea level), MethaneAIR datasets have a 4.5 km swath gridded to 10m x 10m pixels with 17 - 20 ppb standard deviation on a flat scene. MethaneAIR was

5 deployed in the summer of 2021 in the Permian Basin to test the accuracy of the retrieved methane concentrations and emission rates using the algorithms developed for MethaneSAT. We report here point source emissions obtained during a single-blind volume controlled release experiment, using two methods: (1) The modified Integrated Mass Enhancement (mIME) method estimates emission rates using the total mass enhancement of methane in an observed plume combined with winds obtained from Weather Research Forecast driven by High-Resolution Rapid Refresh meteorological data in Large Eddy Simulations

mode (WRF-LES-HRRR). WRF-LES-HRRR simulates winds in stochastic eddy-scale (100 - 1000 m) variability, which is particularly important for low-wind conditions and informing the error budget. The mIME can estimate emission rates of plumes of any size that are detectable by MethaneAIR. (2) The Divergence Integral (DI) method applies Gauss's theorem to estimate the flux divergence fields through a series of closed surfaces enclosing the sources. The set of boxes grows from the upwind side of the plume through the core of each plume and downwind. No selection of inflow concentration, as used in the

mIME, is required. The DI approach can efficiently determine fluxes from large sources and clusters of sources but cannot resolve small point emissions. These methods account for the effects of eddy-scale variation in different ways: the DI averages across many eddies, whereas the mIME re-samples many eddies from the LES simulation. The DI directly uses HRRR winds,

while mIME uses WRF-LES-HRRR wind products. Emissions estimates from both the mIME and DI methods agreed closely with the blinded-volume controlled releases experiments ($N = 21$). The York regression between the estimated emissions and the released emissions has a slope of 0.96 [0.84, 1.08], $R = 0.83$ and $N = 21$, with 30% mean percentage error for the whole data set, which indicates that MethaneAIR can quantify point sources emitting more than 200 kg/hr for the mIME and 500 kg/hr for the DI method. The two methods also agreed on methane emission estimates from various uncontrolled sources in the Permian Basin. The experiment thus demonstrates the powerful potential of the MethaneAIR instrument and suggests that the quantification method should be transferable to MethaneSAT if it meets the design specifications.

## 1 Introduction

Methane ($CH_4$) is the second most important anthropogenic greenhouse gas, with more than 80 times the warming potential of carbon dioxide ($CO_2$) in the first 20 years after its release (Myhre et al., 2013; Etminan et al., 2016). Due to its shorter atmospheric lifetime and higher thermal infrared absorbing efficiency, reduction of methane emissions offers an attractive option for near-term mitigation of greenhouse gas emissions (Shindell et al., 2012). Since the oil and gas (O&G) industry accounts for more than 22% of anthropogenic methane emissions, reducing O&G methane emissions is an effective strategy for reducing greenhouse gas emissions with the added benefit of reduced loss of valuable natural gas (Saunois et al., 2020). Identifying and quantifying methane emissions over large regions with high accuracy is crucial for achieving this goal.

Motivated by these concerns, several remote sensing instruments designed to image high concentrations of methane very close to point sources ("point source imagers") have been introduced in the last decade, including hyperspectral airborne systems such as AVIRIS-NG of Carbon Mapper (Frankenberg et al., 2016) (not explicitly designed for methane measurement), Ball Aerospace' Methane Monitor (Bartholomew et al., 2017), Kairos (Sherwin et al., 2021), Bridger Photonics (Johnson et al., 2021), and satellites including PRISMA (Guanter et al., 2021), Sentinel-2 (Varon et al., 2021), and WorldView-3 (Sánchez-García et al., 2021) (Sentinel-2 or WorldView-3 were also not designed for methane measurements), plus a satellite interferometer, GHGSat (Jervis et al., 2021). There are also global mapping satellites, TROPOspheric Monitoring Instrument (TROPOMI) (Veefkind et al., 2012), with the capability to map the entire globe daily at a much lower spatial resolution (5km x 7km) and The MEthane Remote sensing Lidar missioN (MERLIN) (Ehret et al., 2017) at 200km by 200km resolution. Nevertheless, global mapping satellites, limited by their resolution, cannot detect small methane emissions from O&G operations. Sources too small to detect individually may constitute a significant fraction of all methane emissions in US O&G production (Omara et al., 2022) although the relative contributions of small and large sources to total emissions are uncertain (Sherwin et al., 2023a).

The Environmental Defense Fund (EDF) and its wholly-owned subsidiary, MethaneSAT LLC, have developed a satellite mission, MethaneSAT, to fill the gap between these remote sensing approaches. MethaneAIR is an aircraft-based precursor of MethaneSAT, using very similar spectroscopy, for developing and validating MethaneSAT algorithms (Conway et al., 2023; Staebell et al., 2021). Both instruments were designed to achieve wide spatial coverage with fine spatial resolution and high precision compared to existing remote sensing instruments. MethaneSAT, scheduled for launch in Q1 of 2024, is designed

to have the capability to quantify methane emissions at regional scales (10 km - 100 km), including diffuse emissions as well as detailed resolution of point sources. MethaneAIR, currently deployed, can also provide high-resolution large-spatial scale methane emission detection and quantification as a stand-alone instrument. While the algorithms should be transferable between the two, adjustments will be needed to match MethaneSAT's coarser resolution, higher detection limit, and broader spatial coverage compared to MethaneAIR data (see Table S1 for specification comparisons).

This paper presents the results from the first flights of MethaneAIR, focused on validating emission estimates for O&G sources based on MethaneAIR retrievals. Prior remote sensing studies have introduced several methods to estimate tracer emissions from point sources using methane concentration data. Methods have included applications of Gauss's divergence theorem, including the divergence integral (DI) approach (Frankenberg et al., 2016), mass flux (Conley et al., 2016), or cross-sectional flux methods (Varon et al., 2018). These methods assume that the flux across an enclosing surface around the source equals the integral of flux divergence over the same enclosing surface. The accuracy of these methods is limited by the accuracy of the wind profiles used, the spatial resolution and precision of methane observations, knowledge of the vertical distribution of methane and boundary conditions, and the influence of nearby methane sources. A source pixel method can be used when resolution is coarse (up to at least $10 \times 10 \ km^2$), but measured sensitivity is high (Jacob et al., 2016). This method is based on observations very close to the point emission, allowing the observer to neglect information from the plume downwind where turbulent motions may lead to complex morphology. More recent approaches such as integrated mass enhancement (IME) and machine learning (Varon et al., 2018; Jongaramrungruang et al., 2019) combine the measured excess mass of methane near a source, compared to inflow concentrations, with an effective wind velocity flushing the excess downwind, to obtain an emission estimate (discussed in detail below). These methods work best with a fine spatial resolution ($\leq 1$ km) and high measurement sensitivity (column precision $\leq 10 \ \%$). However, cases with multiple point sources in close proximity to one another are still challenging for all the methods.

We carried out a single-blind volume-controlled release experiment to verify the methods for quantifying methane concentrations and point source emissions using MethaneAIR data. The ground team (the Stanford team) released methane at various metered emission rates from a site near Midland, TX, without sharing the information with the MethaneAIR team. The MethaneAIR team repeatedly flew over the site on two different days, and estimated emission rates of methane without prior information on actual emission rates or measured wind speeds, and submitted the results to the Stanford team. The Stanford team then revealed the volume-controlled release rates. The MethaneAIR team then made minor adjustments to their retrieval algorithms and quality control framework ("decision tree"). This procedure is similar to that used in prior remote sensing blinded-volume controlled release efforts including Sherwin et al. (2021), Frankenberg et al. (2016), Sherwin et al. (2022), and Johnson et al. (2021).

Our methods build upon previous IME and DI algorithms that we adapted to take advantage of MethaneAIR's high resolution and large spatial coverage. We used the Weather Research Forecast model version 3.9.1 in Large Eddy Simulation mode combined with geographical and mesoscale data (HRRR meteorological fields, see below) to simulate eddy-scale winds and vertical mixing. Our application of a high-resolution meteorological (LES) model provides important information for source estimation at the scales measured by MethaneAIR. We validated our results by (1) using the single blinded-volume controlled

release experiment and (2) comparing our modified integrated mass enhancement (mIME) and DI methods when applied to uncontrolled sources in the Permian Basin. Our DI method also introduces new features to the application of the divergence integral adapted for our airborne mission, as detailed later in this paper.

## 2 Data and Methods

### 2.1 MethaneAIR Research Flights

In addition to the controlled-volume experiments, we report here MethaneAIR research flights focused on the Permian Basin, straddling the Texas-New Mexico border, and the nearby Midland-Odessa areas where O&G activities are abundant (see Table 1). Methane from nearby sources can confound the identification and quantification of emissions from a particular source, and data from these flights helps us to understand the impact of adjacent sources on our quantification of emissions from point sources in the real world. All the MethaneAIR flights originated from the Rocky Mountain Metropolitan Airport in Broomfield, CO, using the NSF/NCAR GV HIAPER Aircraft (UCAR/NCAR - Earth Observing Laboratory, 2005). Research flights RF04 and RF05 targeted the volume-controlled releases, plus validation to an EM27/SUN solar viewing spectrometer in East Colorado. The plane repeated overflights of the release site from around 12960 m altitude, at approximately 20-minute intervals, for up to 6 hours while the crew on the ground manipulated the release rates. Methane emitted from nearby O&G activity interfered with the observations on at least 3 overpasses. The first three research fights (RF01 - RF03) were successful engineering fights yet omitted from the paper since our main purposes were for instrument functionality confirmation rather than methane data collection. Here, we show the results of 9 and 12 cloud- and interference-free images for RF04 and RF05, respectively. The flight tracks can be found in the supplement (FigureS32, FigureS33)

Flights RF06 and RF07 aimed to map the Delaware sub-basin of the Permian oil and gas field. The strategy for mapping the areas of interest is to create repeated tracks over the target areas with partial overlaps (i.e., area mapping). The overlapped segments act as buffers of missing data, allowing us to observe the changes in the column-averaged dry-air mole fraction of methane ($XCH_4$) over time.

**Table 1.** Summary of research flights of interest during the MethaneAIR campaign over the summer of 2021

| Research Flights | Dates | Targets |
|---|---|---|
| RF04 | 30 July 2021 | Blinded-volume controlled releases & EM27/SUN |
| RF05 | 3 August 2021 | Blinded-volume controlled releases & EM27/SUN |
| RF06 | 6 August 2021 | Delaware Sub-basin |
| RF07 | 9 August 2021 | Delaware & Midland Sub-basins |

During the survey of the Delaware Sub-basin of the Permian Basin flight (RF06), MethaneAIR covered more than 9,000 square kilometers in 2.2 hours. An example of what MethaneAIR observed from a source of methane is shown in Figure 1 (a). This source was measured during RF06 and identified as the MiVida Gas Processing Plant in Barstow, TX. When the flight

tracks are overlapped, we can combine all the data segments and create a mosaic map of the entire area of interest as shown in Figure 1 (b).

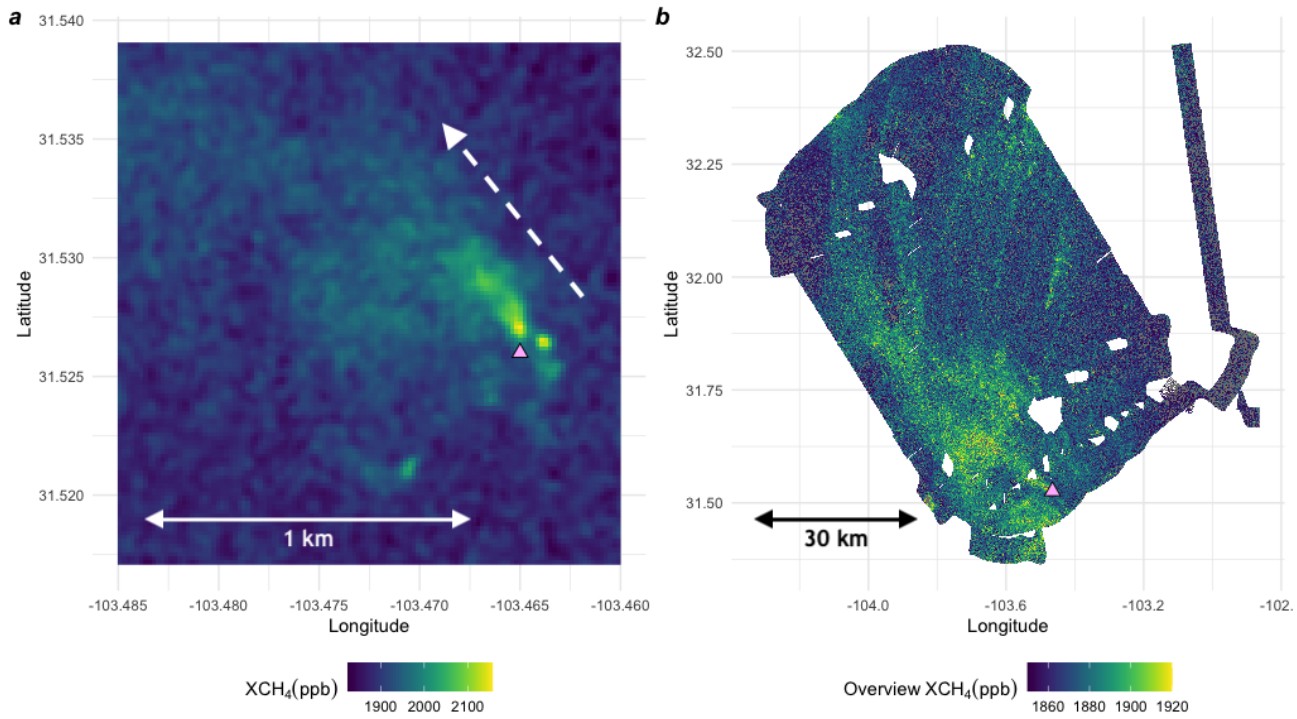

**Figure 1.** Examples of the MethaneAIR data on 6 August 2021 (a) MiVida Gas Processing Plant (pink triangle) from RF06 segment 10 (approximately 2km x 2km image) and (b) mosaic image (75 km x 120 km) of the entire RF06 scene from all the segments including the segment shown in Figure 1. The prominent gap areas in this picture were not imaged during the overlapping turns of the pattern. The white dashed line indicates the approximate wind direction at the processing plant site. Full flight track information can be found in S6.

In 2022, we conducted another single-blind volume-controlled experiment over two research flights near Phoenix, AZ. Only RF01E and RF03E research flights that include the experiment are shown here. The flight tracks can be found in the supplement (FigureS34, FigureS35). We obtained results of 11 and 13 cloud- and interference-free images for RF01E and RF03E, respectively.

**Table 2.** Summary of research flights of interest during the MethaneAIR campaign over the fall of 2022

| Research Flights | Dates | Targets |
|---|---|---|
| RF01E | 25 October 2022 | Blinded-volume controlled releases & EM27/SUN |
| RF03E | 29 October 2022 | Blinded-volume controlled releases & EM27/SUN |

## 2.2 MethaneAIR Data Retrievals

From our cruise altitude of 12960 m above ground ((13850 above sea level), the across track distance between pixel centers is 5 m and swath of about 4.5 km or 863 pixels. The along-track sampling has centers ∼25 m apart. The MethaneAIR $XCH_4$ data were retrieved from the 1.65-micron band, with $CO_2$ as the retrieval proxy for the optical path, using the Smithsonian PLanetary ATmosphere interface to VLIDORT (SPLAT-VLIDORT) radiative transfer code (Chan Miller et al., 2023). We use $CH_4$, $CO_2$, and $H_2O$ prior profiles from the TCCON GGG2020 website for hte retrieval (Toon, 2022a, b; Wunch et al., 2011). Our multilayer methane retrieval algorithm is based on the $CO_2$ proxy method. We use true methane columns comprising the paths from space to the ground and the ground to the aircraft ($\approx 12$ km). We account for the influence that topographic variations will have due to the profiles of methane and carbon dioxide using the averaging kernel for our sensor. The tropopause height variations may influence the methane columns due to less methane in the stratosphere. The influence may play a role at some MethaneAIR altitudes and will certainly affect the MethaneSAT data since the satellite will be higher up in the atmosphere. Since most excess methane is within the boundary layer (below the aircraft), we assume the averaging kernel for anomalies applies to the lowest kilometers. The averaging kernel is slightly larger near the surface than in the upper atmosphere.

MethaneAIR utilizes two Offner spectrometers (Headwall Photonics) covering specific wavelength ranges and InGaAs detectors (Princeton IR Technologies). We employ the $CO_2$ retrieval proxy method in the 1.65 $\mu$m $CO_2$ band and the $2\nu_3$ $CH_4$ band. The $CO_2$ proxy is known for its robustness under moderate aerosol conditions (Frankenberg et al., 2005, 2006; Butz et al., 2010; Parker et al., 2015), with the main errors due to surface albedo differences between the $CO_2/CH_4$ fit windows (Parker et al., 2020). Large aerosol concentrations will be flagged by the cloud routines. The main sensitivity to aerosols in the proxy approach is associated with differences between the albedo in the $CO_2/CH_4$ bands, which will manifest as a correlation with surface albedo. Individual scenes can be analyzed to remove this.

Based on the engineering flights, the albedo dependence for MethaneAIR data is lower than anticipated, given that during the campaign, observations were often over regions blanketed by haze from long-range transport of smoke from fires in the Western United States and Canada. The size distribution may be small enough for the aerosol optical depth to be insignificant at 1600 nm since a large fraction may be in drier air in the free troposphere. The sign and magnitude of albedo bias will be scene-dependent, determined by the prior profile bias and additional light-path modifications induced by aerosol scattering. The detailed discussion can be found in Chan Miller et al. (2023).

The $XCH_4$ data were later projected onto a 10 m × 10 m grid; in our report to the blinded experiment, we used a nearest-neighbor gridding method that accounts for sensor oversampling on the ground (Appendix S2.4 ), whereas in our post-unblinding ("best") rendering, we used the "snowflake" gridding approach of Sun et al. (Sun et al., 2018). The gridded

images were denoised using a 2-dimensional mass conserving Gaussian filter with a half-width at half maximum of 0.85 pixels, yielding denoised $XCH_4$ 10m × 10m gridded data with the noise of 40 ppb ($1\sigma$) for the blinded report and 15-20 ppb ($1\sigma$) for the best rendering. The spatial decorrelation length for the image is ~70m.

## 2.3 Methane Point Source Emission Quantification

We developed two main methane point source quantification approaches, mIME and DI, adapted from the literature for application to MethaneAIR data. To test the accuracy of our quantification methods, independent of a plume-finding exercise, we assumed that the source location was already identified. We note that, by assuming that source locations were identified, we have a lower chance of having false positives.

### 2.3.1 Large-Eddy Simulation Based Approaches

We used the Weather Research Forecast Model driven by High-Resolution Rapid Refresh (HRRR) meteorological data in the Large-Eddy Simulation mode (WRF-LES-HRRR) to simulate plumes at the source point. The simulated plumes exhibit stochastic behavior in response to eddy-scale winds associated with boundary layer turbulence, providing quantitative information on the uncertainty due to these processes. The WRF-LES code we used was first developed to simulate $CO_2$ transport in the US Upper Midwest and Indianapolis areas (Lauvaux et al., 2012; Gaudet et al., 2017). Our WRF-LES model can take
GRIB2 files of the meteorological data of interest (e.g., HRRR, European Centre for Medium-Range Weather Forecasts, and Global Forecast System data) and static geographical data as inputs. For this study, we used HRRR winds to drive the WRF-LES combined with the highest resolution static geographical data available on the official WRF website (Blaylock et al., 2017; Skamarock et al., 2008). Given the knowledge of point source locations, we nested (one-way) the domains starting from the native HRRR resolution of 3 km × 3 km to 111 m × 111 m grid cells by nesting the domains three times at 3:1 ratio each
time. The innermost domain is 103 pixels by 103 pixels or 11.44 km by 11.44 km. There are 49 vertical layers, with the highest resolution near the surface and coarser towards the top of the atmosphere. We used the Noah Land-Surface Model scheme for the WRF surface option, 1.5 order TKE closure (3D) for the eddy coefficient option of the innermost domain, and Mellor–Yamada– Nakanishi–Niino (MYNN)-Eddy Diffusivity-Mass Flux (EDMF) scheme (Nakanishi and Niino, 2006, 2009). The closure in the boundary layer limits effective eddy resolution to 50 m – 100 m; by limiting our grid to 111 m, the run time
for the model is tractable, and the simulation of the plumes is realistic on a spatial scale approaching MethaneAIR resolution. We ran the simulations for the days of interest, starting 5 hours before the first observation on each day and ending half an hour after the last observation of interest.

The center of the innermost domain was set to align with the emission source location. When multiple sources are clustered in close proximity, within plus or minus 1/3 the size of the domain from the center (approximately 3.5 km for most of our
innermost domains), those sources can be placed in the same model run without expanding the size of the domain. If the sources spread out more than 1/3 of the domain size, we can expand the domain sizes accordingly with additional computation time. Our WRF-LES-HRRR writes innermost domain outputs every minute. The outputs include all the model parameters and a specific methane concentration for every point in the domain.

We identified and isolated the plume from each point source by defining a threshold above inflow concentration, 1.5 standard deviations of the inflow values, above the median value of the inflow, into the source region (see Figure 2B for an example of an isolated plume and Figure S10 for an example of emission rates as a function of thresholds), setting values below that values to NA. Since we have full knowledge of the emission rate underlying the plume from each LES scene, we could, in principle, determine the "effective wind speed" ($U_{\text{eff}}$, the turnover rate for the mass in the plume; Varon et al. (2018)) that would give the best fit to the observed plume shape for each encounter, and then estimate the emission rate from the ratio of the mass of the observed plume to the mass of the LES plumes times the nominal emission rate given to the LES model. We used this method, called the Ratio Method (Appendix S2.3), in Irakulis-Loitxate et al. (2021) where the resolution of the plume images was close to the LES resolution. However, here the LES spatial resolution is coarser than MethaneAIR resolution, so we modified the Integrated Mass Enhancement (IME) method of Varon et al. (2018) to adapt to our conditions, as described below.

**Modified Integrated Mass Enhancement**

The concept of *IME* emission started with the computation of the total mass enhancement in a defined plume. The *IME* is the integrated mass enhancement of the plume and defined as

$$\text{IME} = \sum_{j=1}^{N} \Delta\Omega_j A_j. \tag{1}$$

where $\Delta\Omega_j$ is the mass enhancement per pixel, defined as the total column of methane enhancement relative to the inflow concentration and $A_j$ is the area of each pixel $j$. To obtain the emission rate, we introduce the following equation

$$Q = \frac{U_{\text{eff}}}{L} IME, \tag{2}$$

where $Q$ is the emission (kg/hr), $L$ is the scale length defined as the $\sqrt{A}$, and $A$ is the area of the plume ($m^2$). We multiply the *IME* by the turnover rate ($U_{\text{eff}}$ /L). The effective wind speed $U_{\text{eff}}$ is a function of $\log(U_{10}) + 0.6$, where $U_{10}$ is the wind speed at the 10m height.

The Varon et al. (2018) IME method used $U_{10}$ from operational meteorological products, whereas our modified approach (mIME) uses the 10 m root-mean-square wind from each LES realization specifically run for the case of interest, averaged over $\pm$ 15 minutes from the sampling time. The relationship between $U_{10}$ and $U_{\text{eff}}$ is given by a set of empirical coefficients (Appendix S2.1; (Varon et al., 2018)), reflecting the similarity of the root-mean-square winds in our LES runs to the winds from the large ensemble of (idealized) LES simulations in Varon et al. (2018). Since our LES winds simulate the eddy-scale component applicable to each plume, our simulations give us an estimate of the variability of the IME at the time of each observation, providing the uncertainty for each event.

### 2.3.2 Divergence Integral Method

The DI method is based on Gauss' divergence theorem applied to surfaces enclosing the source. The surface integral is estimated based on XCH$_4$ measurements along a rectangle surrounding the source following Eq. S5 and Eq. S6:

$$\Phi_{surf} = \overset{\text{around rect}}{\sum} \left( XCH_{4_i} - \langle XCH_4 \rangle_{rect.} \right) \cdot n_{\text{column}} \cdot M_{CH_4} \cdot v_{\text{perpendicular}} \cdot \Delta l + \left\langle \frac{\partial m}{\partial t} \right\rangle \tag{3}$$

where $\Phi_{surf}$ is the flux (kg/hr) into the enclosed volume from the land surface, $XCH_{4_i}$ is an individual measurement along the rectangle, $\langle XCH_4 \rangle_{rect.}$ is the mean of all measurements along the rectangle, $\Delta l$ is the distance between successive $XCH_4$ measurements, $n_{column}$ is the moles of air in the column based on the surface pressure from HRRR, $M_{CH4}$ is the molar mass of methane, $v_{\text{perpendicular}}$ is the wind speed perpendicular to the sides of the rectangle, and $\left\langle \frac{\partial m}{\partial t} \right\rangle$ is the rate of change of total mass in the enclosed volume ($\sim$0 for plume-size volumes). The wind speed was obtained from HRRR but was rotated to match the wind direction of the observed plume using the difference between the major axis of the moment of inertia and the HRRR wind direction (see Appendix S2.2). Errors in wind direction and wind speed from HRRR will affect the DI method's calculated emissions. We used the wind direction information in the plume image to reduce this potentially large source of error.

The flux was calculated for a series of rectangles ("expanding boxes") with the distance from the source to the downwind edge of the rectangle ranging from approximately 100 m to the length of the observed plume (typically 300 - 2000 m). To fully take advantage of the information in the MethaneAIR image, the size of the rectangle was increased sequentially by one pixel (10 m) in each direction, with the exception of the upwind side, which was increased by a pixel for every fourth rectangle to avoid catching other sources yet still average over the background (Figure 2c).

Figure 2d shows the calculated fluxes around the unlit flare near the controlled release on 3 August 2021 for individual rectangles as a function of the distance from the source to the downwind edge of the rectangle. Boundary layer eddies break up the plume structure, leading to a buildup of methane concentration in some areas and depletion in others. Since these eddies are not resolved by HRRR, we see oscillations of the computed flux values as a function of distance, on the scale of the associated structures. However, by averaging the flux over the range of distances, we arrive at a robust flux estimate for the source that averages over the eddy-scale variability of the winds. The estimated flux for the individual source is the average flux for all rectangles crossing the plume, in this case, rectangles with edge distances from 100–800 m. (*Note*: The blinded DI emission estimate initially reported to Stanford used the alternate name "Gaussian Integral" for this method).

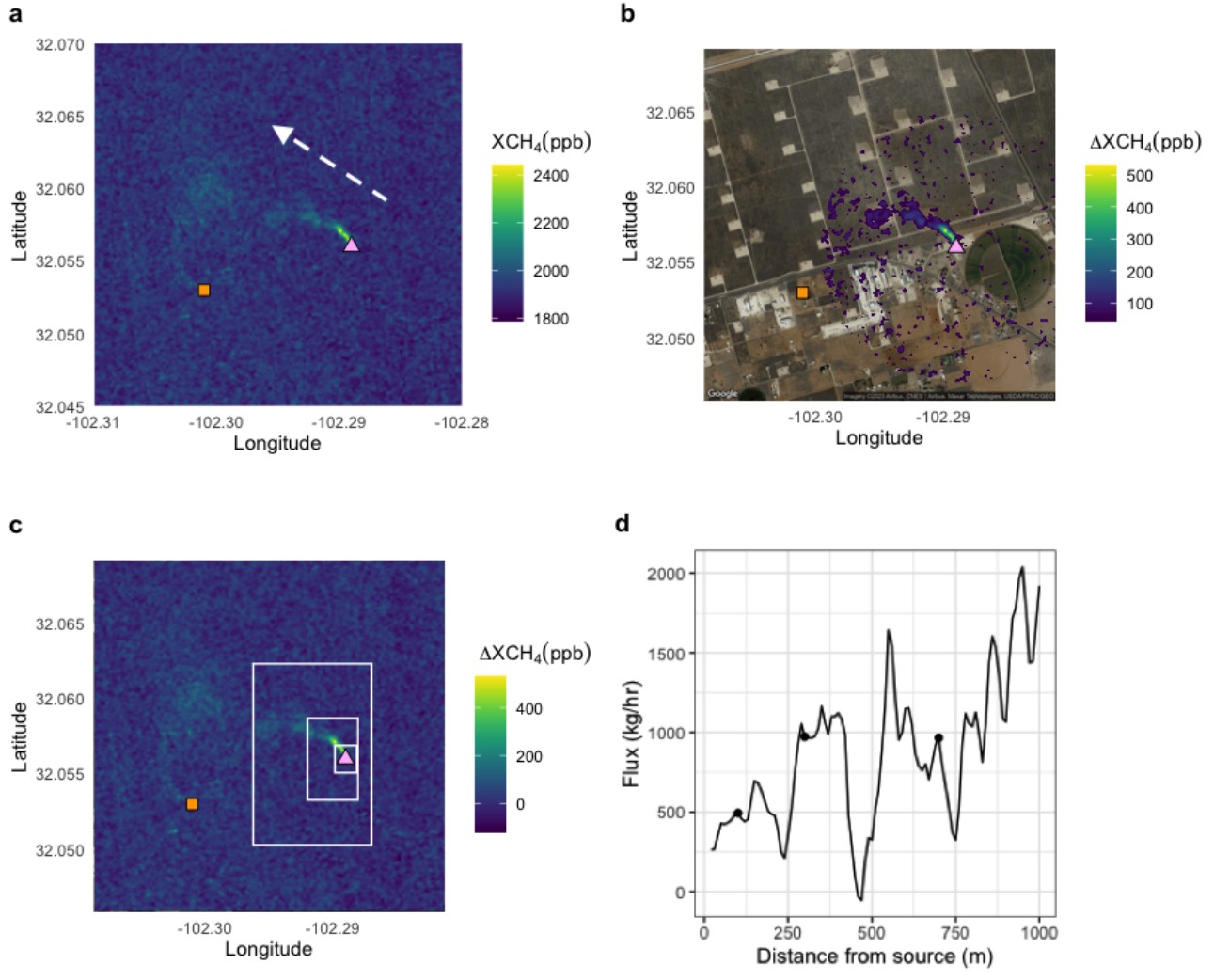

**Figure 2. a**: MethaneAIR image of a blinded volume-controlled release (orange square) and an unlit flare (pink triangle) observed on 3 August 2021. The white dashed line indicates the approximate wind direction at the unlit flare site. **b**: A sample of the isolated unlit flare plume using the mIME method. The isolated plume is superimposed on a ©Google Map 2022 satellite layer. **c**: Three example rectangles are used to calculate the flux divergence. **d**: Calculated flux divergence as a function of distance from the source to the downwind edge of the rectangle. Circles indicate the position of the example rectangles are shown in Figure 2c. The fluctuations of the apparent flux with distance from the source reflect the influence of eddy scale motion as well as contributions of excess methane from nearby sources. The influence of eddy-scale motions is evident in the oscillation at approximately 250 m intervals, the apparent length scale for eddies at this overpass. The surface flux is estimated by averaging the flux divergence over several eddy scales to average out $\partial m/\partial t$. In this example, we averaged the DI from $80 - 700$m to avoid influence from other sources nearby that increase the DI beyond 700m from the source.

## 2.4 Wind Validation

Since the accuracy of the winds directly affects the emission estimates, we examined other sources of wind data to assess the associated uncertainty. We compared our modeled WRF-LES-HRRR winds to wind observations from nearby airports as well as ground-based wind measurements from an instrument at the controlled release site. The comparisons suggest that WRF-LES-HRRR winds are comparable to ground-based wind measurements near the site while the larger scale or nearby airport winds generated larger errors.

### 2.4.1 Ground Wind Validation

Wind speeds at 10 meters above ground level was measured by a Gill Instruments WindSonic 60 2-dimensional ultrasonic anemometer. The anemometer was placed on a 4 m pole-mounted tripod, secured to the steel flooring of a JLG 400 s telescopic boom lift bucket. The boom lift bucket is designed for a vertical lift while keeping the platform level, and fine adjustments to its height could be made. The anemometer was placed roughly 40 m S-SE from the releasing stack. The data were withheld from the MethaneAIR team before the first data comparison. We later used measured winds from this site to check our estimations from WRF-LES-HRRR.

### 2.4.2 Automated Surface Observing Systems Wind Validation

Automated Surface Observing Systems (ASOS) provide meteorological measurements at airports in the United States, including wind surface observations. The data are reported continuously every minute and released to the public within two weeks (US Department of Commerce). ASOS winds can be used as quality checks for wind products used in MethaneSAT or MethaneAIR emission estimates when ground-based measurements at the source are not available. The ASOS resolution is 1 kt or 0.5 meter per second, limiting its utility under low wind conditions.

We compared the ASOS data from nearby airports to the average WRF-LES-HRRR winds near the source. The time series of the winds from different products are shown in Figure S8. The purple crosses represent ASOS winds from the Midland International Air & Space Port (MAF). Because the locations were slightly different, we only look at the variation of the winds within the window of interest. When the variations of the ASOS winds and the WRF-LES-HRRR winds are from the same distribution (two-sided t-test p-value is greater than 0.05), the WRF-LES-HRRR winds are trusted. For the days of blinded-volume controlled release experiments, all the p-values were greater than 0.05 and the LES winds were used in all cases.

## 2.5 Error Analysis

Since resampling the same view of a plume multiple times is not feasible due to the changing meteorological conditions and flow rates, we used the Monte Carlo simulation principle to calculate the confidence intervals of the mIME estimates. Based on the original observation, we generated a thousand synthetic observations. To simulate a synthetic observation, we sampled $XCH_4$ pixels of the observed inflow with replacement $n$ times, where $n$ is the number of the original inflow pixels. The new resampled inflow mean and standard deviation values were used as a new background and a threshold. After applying the new

threshold to the field of $XCH_4$, we constrained the extent of the new plume using the original area. A new mass was calculated from the new plume. For the winds, we randomized the mean inflow winds by sampling from seven LES snapshots, taken at 5-minute intervals within 30 minutes of the retrieval. We calculated the mIME estimate for each resampled set of values. Finally, we calculated the confidence interval (the 2.5[th] and 97.5[th] percentile of the emission estimates) from the 1000 resampling trials. Our mIME confidence intervals, therefore, include the variances from the LES winds, instrument, and measurement errors for this particular set of MethaneAIR determination of emission rates.

For the DI method, we used a t-test on the set of fluxes obtained for boxes of different sizes expanding in the downwind direction from the source to estimate the 95% confidence interval of the fluxes. Schneising et al. (2020) used a similar approach but only varied the position of the downwind side of the rectangle, leaving the other sides fixed. By changing the position of all sides of the rectangle, we average over different clean segments and reduce errors based on concentration fluctuations outside of the plume (unlike our application, Schneising et al. (2020) were able to average over a longer time series enabling them to average out this uncertainty, but we have only one snapshot). We use this method on a much different length scale, as Schneising et al. calculated basin-level emissions, on the scale of 400 km, compared to our goal of quantifying individual emitters with plumes spanning 0.4 to 10 km.

## 2.6 Targeted Emissions Sources

During the MethaneAIR campaign in the Permian over the summer of 2021, we applied mIME and DI approaches to obtain the emission rates, validated the winds, and estimated the errors for the blinded volume-controlled releases of methane and other uncontrolled methane sources in the area.

### 2.6.1 Blinded-Volume Controlled Release Experiment

Controlled release testing was conducted to evaluate the quantitative performance of MethaneAIR. We conducted repeated passes over the metered methane point source during controlled release intervals. The MethaneAIR team was aware of the release location but not of the release rate of the methane source nor whether any release was happening at a given overpass. Stanford University organized the controlled release experimental campaign (32.053 °N, 102.301 °W) near Gardendale, Texas, in July-August 2021, sampled on two days of the campaign. A suite of other teams in controlled release testing, including multiple aerial teams (Rutherford et al., 2023) and satellite teams (Sherwin et al., 2023b). MethaneAIR participated in the campaign on July 30 and August 3, 2021.

MethaneAIR was in an earlier stage of development than other tested systems, having never obtained any scientific data prior to the sampling controlled release of methane (limited engineering data had been obtained in 2019). Due to this novelty, a slightly modified testing agreement was created with a collaborative publication plan in lieu of independent publication results by the Stanford team, as was performed for other teams. Other protocol changes include: (1) MethaneAIR submitted three sets of estimates using different algorithms and selected which was considered to be best, before unblinding (2) MethaneAIR requested that information about wind direction and the presence or absence of meaningful wind speeds be provided, which

Stanford sent to Harvard during the experiment seven times. This information was intended solely to inform the aircraft's flight trajectory, which was not in fact modified from the filed flight plan during the testing.

The limitations of this experiment include (1) a small range and low flow rates of the controlled release values (all less than 1000 kg/hr) compared to larger than 1000 kg/hr releases observed by other participants, (2) the release sites were shifted between 3 m and 5.25 m altitude and (3) the number of blinded-volume controlled releases measured was limited to 21 points on two days. The experiment was, however, fully blinded — the metered emission values were not revealed to the MethaneAIR team until after initial emission estimates from MethaneAIR using the mIME and DI methods were reported to the Stanford team, including all overpasses where the IME method detected an $XCH_4$ plume above a 1.5 standard deviation threshold within one kilometer from the releasing site. An unplanned limitation was imposed by significant methane emissions from sources close to the releasing point, including an unlit flare with a high emission rate located at 32.056°N, 102.288°W, ~1 km to the east. This suggests that sources within 1 km may impact the ability to quantify emissions. Emissions from this source were detected on all passes of the blinded-volume controlled release experiment. MethaneAIR has higher sensitivity and a large pixel size (10m x 10m or 20m x 20m, depending on the gridding choices) than other airborne sensors. The methane from this and other extraneous emission sources sometimes interfered with MethaneAIR observations, especially for the DI.

After unblinding of the 2021 experiment, we observed another set of single-blind volume controlled releases near Pheonix, AZ (32.821 °N, 111.786 °W) on 25 & 29 October 2022. Stanford's 2022 instruments have lower uncertainty than those used in the 2021 experiment (El Abbadi et al., 2023). With better precision in the instruments, the Stanford team reported the 2022 emission rates as 60-second means instead of 90-second means in 2021.

### 2.6.2 Emission Rates for Miscellaneous Methane Sources in the Delaware Basin

Many miscellaneous methane sources were detected during the MethaneAIR research flights RF06 and RF07. We compared the two emission estimates, mIME and DI, and report here both values. We manually identified the scenes with high methane concentrations and co-located with any O&G infrastructures or reported leaks. The sources from RF06 and RF07 enabled us to estimate methane emissions from 9 overpasses of the MiVida gas plant (31.524°N, 103.467°W, Bartow, TX) over two days. We discovered and quantified a very large pipeline leak in New Mexico on RF07.

## 3 Results and Discussion

### 3.1 Blinded Volume-Controlled Release Experiment

For two testing days, MethaneAIR collected 21 data points. We included emission estimates based on the mIME, the DI, and the Ratio methods presented in Table S4 as the original results submitted to Stanford by MethaneAIR on 1 February 2022 (before unblinding the metered release volumes). Based on the decision tree workflow in Appendix S4.4 using both mIME and DI results, a set of blinded best-estimate emissions was submitted to Stanford as the original submission. In this original blinded submission, one emission estimate of the 21 was made using the mean of DI and mIME, and the rest were made using mIME

only. After the initial report of the blinded best-estimate data, we decided to use only the mIME method for the consistency of all the data points. Emission rates were not modified post-unblinding. However, some edge cases were flagged via the decision tree. Both blinded best estimate and post-unblinding results are presented in Table S2. Only post-unblinding mIME results are presented in Figure 3. For references, the post-unblinding DI results are presented in Table S3. The post-unblinding mIME and DI results are also presented in Figure S1 and Figure S2 as bar plots representing the emissions and color-coded following the decision tree. In addition, the post-unblinding DI estimates are presented in Figure S4 as an analogous plot of Figure 3 similar to the blinded best-estimate results that are presented in Figure S3.

After the Stanford team provided the true emission rates from the experiment, we designed an algorithm that could reliably reject mIME estimates that were false positives or below the detection threshold for a point source. We also developed a more comprehensive bootstrapping approach to account for uncertainty from background concentrations and thresholding in addition to the winds. In the second set of results, this filtering algorithm designated "below detection limit" six emission passes from the dataset; the quantified emissions were slightly changed after unblinding due to the new bootstrapping results. One flagged data point: MethaneAIR estimated the emissions to be above 200 kg/hr when the metered emission is below 200 kg/hr; this observation suggested that, when emissions are below the detection limit, the estimated emissions are uncertain and can be overestimated. These flagged and no-detection data points were not removed before we applied the OLS or York regressions. Based on this comparison, we determined that the detection limit for the MethaneAIR 1x1 retrievals using the mIME method was approximately 200 kg/hr. This value is preliminary because it is based on a small sample size, for two days, in contaminated areas; future work is needed to fully characterize the detection limit (such as using methods presented in Conrad et al. (2022)).

Figure 3 shows methane flux rates determined by the MethaneAIR mIME method compared to the controlled release rates from the Stanford team from RF04 and RF05 flights on 30 July and 3 August 2021, respectively (all the plumes identified and used here are shown in Appendix S5), including the minor adjustments after unblinding and designating with red circles the false positives in the blinded submission. The red square designates the false negative in the blinded submission. The York regression slope (Appendix S2.5, MethaneAIR (y-axis) vs blinded-volume controlled releases (x-axis), is 0.96 ($\pm$ 0.12, 95% confidence interval), using error estimates from our bootstrap (y-axis, mIME emission rates) and from the Stanford team (x-axis, reported metered flows). The intercept is 88 kg/hr ($\pm$ 27 kg/hr). Since the uncertainties in the controlled release rates are not negligible, the York slope provides the maximum likelihood estimate of the relationship between controlled release and emissions obtained from analyzing MethaneAIR data. We compare this fit to the ordinary least square (OLS) regression results, often used to analyze this type of experiment, where the slope is 0.85 ($\pm$ 0.13, 95% confidence interval), $R$ of 0.83, and the intercept is 113 kg/hr ($\pm$ 45 kg/hr). The limitations of this experiment include (1) a small range of blinded-volume controlled release values (all less than 1000 kg/hr) and (2) a limited number of blinded volume-controlled releases measured ($N = 21$). The excellent agreement between the fluxes calculated by applying the mIME method to MethaneAIR data and the true controlled release rates provide strong support for the validity of our $XCH_4$ retrievals and flux calculations, subject to those limitations.

**Figure 3.** Volume-controlled release experiment results after unblinding. The results presented in this figure are based entirely on the mIME method. The black circles represent the post-unblinding estimates plotting against the reported metered emissions. The red square represents no detection. The red circles represent data flagged as below the detection limit (S11). The blue solid line is the post-unblinding York fit. The blue shaded area represents a 95% confidence interval of the York fit from the resamples. The orange is the Ordinary Least Squares (OLS) fit. The p-value of 0.075 from the paired t-test between the estimated emissions and Stanford emissions suggests that we cannot reject the null hypothesis that the population mean of the differences is 0 with a 95% confidence interval.

We conducted paired sample t-tests to determine whether the mean difference between the reported emissions from the Stanford team and the estimated emissions is different than zero. Our null hypothesis is that there is no difference between the mean observed and estimated emissions. The test was performed three times for three sets of estimated emissions based on three different wind products: WRF-LES-HRRR winds, HRRR winds, and measured winds at the release site (See Figure S9 for the comparison). The p-values are 0.17, 0.041, and 0.31, respectively. We cannot reject the null hypothesis of no true mean difference between the reported emissions and the estimated emissions derived from WRF-LES-HRR or observed winds. However, the 0.0408 p-value for HRRR winds suggests that there is likely to be a true mean difference between the reported emissions and the estimated emissions. We infer that the WRF-LES simulated winds improve the mIME method compared to simply using HRRR winds in the IME function. Within our limited data, the mIME method works equally well using WRF-LES-HRRR winds or measured local winds.

For two testing days in 2022, MethaneAIR collected 24 data points. We included emission estimates based on both the mIME, the DI, and the best estimate (the average between the two methods) as results submitted to Stanford by MethaneAIR on 22 March 2023 (prior to unblinding the metered release volumes).

Figure 4 shows methane flux rates determined by the average of the MethaneAIR mIME and DI methods compared to the controlled release rates from the Stanford team from RF01E and RF03E flights on 25 and 29 October 2022, respectively, including the low-quality estimates designating with red circles in the blinded submission. As a result, the confidence intervals are much larger than the confidence interval in 2021, which was exclusively determined by the mIME method.

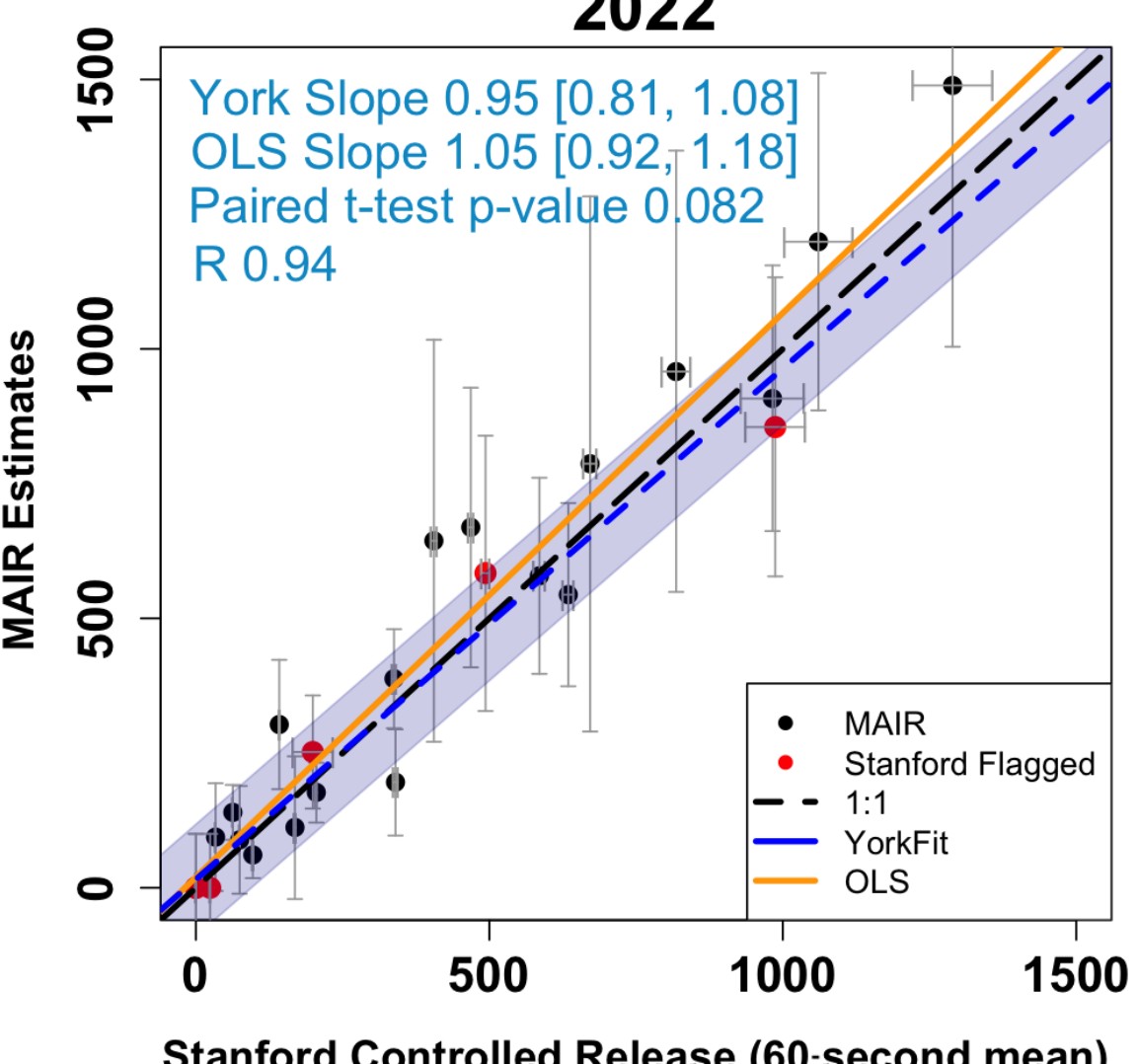

**Figure 4.** Volume-controlled release experiment results after unblinding. The results presented in this figure are based on the average between the mIME and the DI method. The black circles represent the post-unblinding estimates plotting against the reported metered emissions. The red circles represent data flagged as low-quality data points by the Stanford team. None of these data points are flagged by MethaneAIR. The blue solid line is the post-unblinding York fit. The blue shaded area represents a 95% confidence interval of the York fit from the resamples. The orange is the Ordinary Least Squares (OLS) fit. The p-value of 0.082 from the paired t-test between the estimated emissions and Stanford emissions suggests that we cannot reject the null hypothesis that the population mean of the differences is 0 with a 95% confidence interval.

### 3.2 Miscellaneous Methane Sources

#### 3.2.1 Unlit Flare

MethaneAIR observed the unlit flare near the release site 21 times with repeated overpasses on 30 July and 3 August 2021. As shown in Figure S6, the flare emission rates ranged from 500 kg/hr to 2000 kg/hr as calculated by the mIME and DI methods. Based on this comparison and the comparison with the controlled releases, we estimated the detection limit of the DI method to be approximately 500 kg/hr.

#### 3.2.2 MiVida Gas Processing Plant

We quantified emissions from a large consistent emitter, the MiVida Gas Processing Plant, in Barstow, TX, 9 times with repeated overpasses on 6 and 9 August 2021. Typical emission rates were 2000 kg/hr with 30% uncertainty estimate based on the controlled releases experiment results (Figure S7). Several emission points could be identified within the boundaries of this plant.

#### 3.2.3 The Pipeline Leak in New Mexico

We discovered a large pipeline leak at 103.697°W, 32.366°N in New Mexico. MethaneAIR detected the leak on 9 August 2021, not present on the earlier flight on 6 August 2021. Using the mIME and DI methods, we estimate the emission rate to be approximately 5000 kg/hr with 30% uncertainty estimate based on the controlled releases experiment results (Figure S7).

We attributed the leak to a gathering pipeline emission event at the location, confirmed by emission incident reports to the New Mexico Oil Conservation Division on August 26, 2021, which identified the cause as a rupture at a weld along a gathering pipeline segment (?). Based on this report, we estimated a methane emission rate of 8200 kg/h over an 18-hour reporting period, assuming 80% methane content in the gathered natural gas. With simple extrapolation of the lower estimate of 5000 kg/hr from the flight on August 9 until the leak was fixed on August 24, the total methane emission could be more than $1.8 \times 10^6$ kg (1,800 metric tons) over the 15 days.

### 3.3 Confidence Levels of the Emission Estimates

Beyond the confidence flags from the decision tree (Appendix S4.4) and attempts to understand the methane contamination of the scenes of interest by methane from nearby sources (Appendix S4.5), we also evaluated the validity of our emission estimates by plotting the methane emission rates from both methods from all the sources within the same plot (Figure 5). This agreement between the two methods has a slope of 1.30 [1.12, 1.49], $R = 0.94$, and $N = 42$ for emissions exceeding 500 kg/hr. The t-test p-value of the two methods is greater than 0.05 and suggests that the estimates from these two methods came from the same distribution. Since the two methods agree for the blinded-volume controlled releases and the flare sources when the estimated emission rates are greater than 500 kg/hr, we infer that both methods provide valid emission estimates when emission

rates are greater than 500 kg/hr. For rates below 500 kg/hr, both methods detect the sources down to 250 kg/hr, but the DI overestimates the rates due to the influence of nearby sources.

    The mean percentage error of the methods should also be similar to those from the blinded-volume controlled release experiment (around 30%), given that uncertainty from both methods came largely from the influence of eddy-scale variability. The mIME simulation captures the variation over time in the simulations. The DI averages the flux information across various

distances, capturing spatial variability due to eddies.

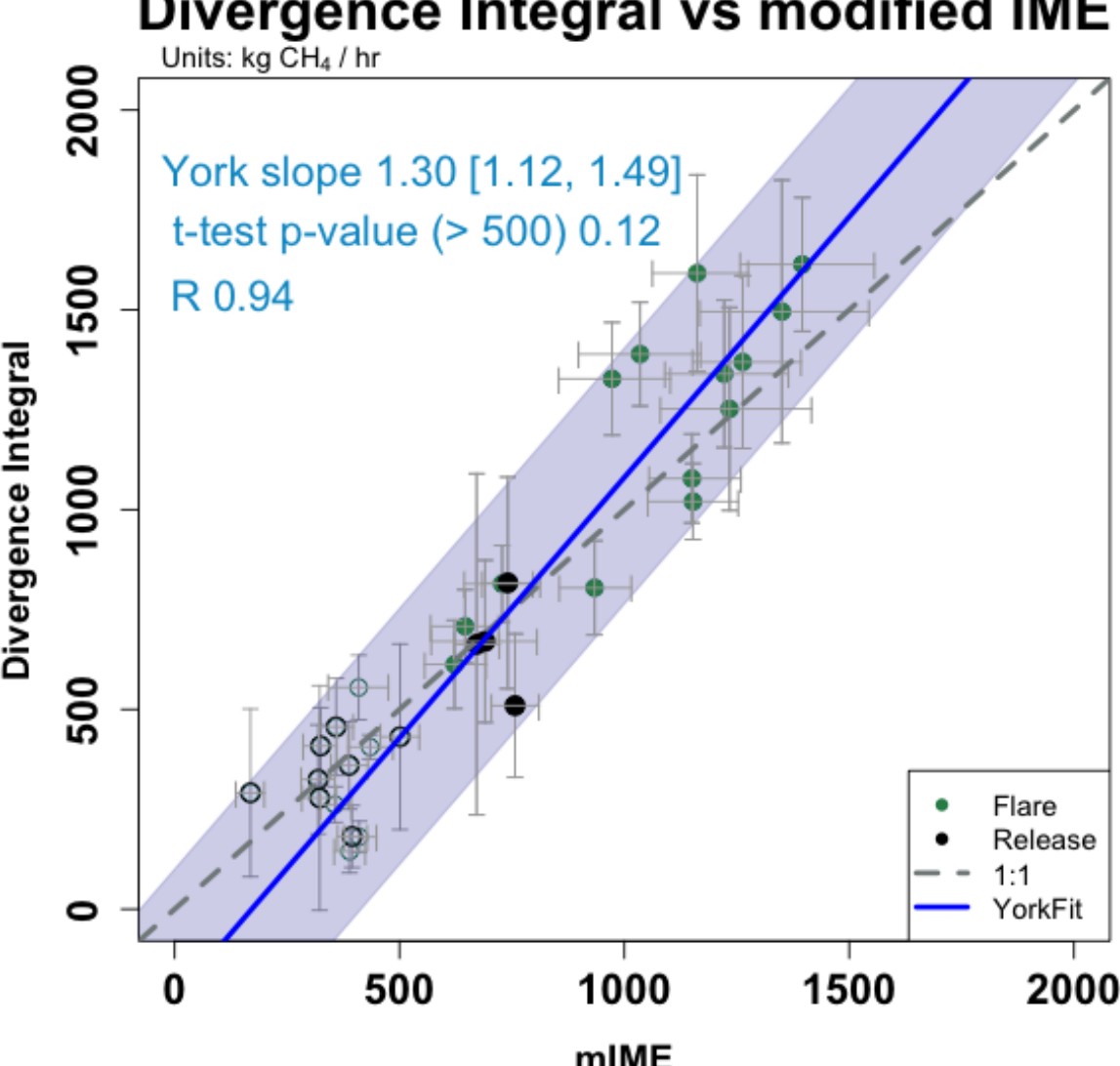

**Figure 5.** mIME vs DI estimated emissions for all valid cases from all the research flights of interest in 2021. Using the algorithms presented here and meteorological fields from HRRR, MethaneAIR observations of $XCH_4$ enabled accurate estimation of methane emission from point sources within 30% errors, close to zero bias, and a detection limit of 200 kg/hr. The black circles represent the blind volume controlled release mIME estimates plotting against the DI emission estimates. The green circles represent the unlit flare mIME estimates plotting against the DI emission estimates. The circles representing emission estimates under 500 kg/hr are open. The circles representing emission estimates over 500 kg/hr are solid. The blue solid line is the post-unblinding York fit. The blue shaded area represents a 95% confidence interval of the York fit from resampling. The paired t-test between the Divergence Integral and the mIME emissions (p = 0.12, rate > 500 kg/hr) indicates no statistically significant difference between the population means for the two methods.

## 4 Conclusions

Our estimates of methane emissions from the blinded volume-controlled release experiment agreed well with the metered flow rates. The 2021 comparison between the MethaneAIR estimates and the known rates provides a slope of 0.96 [0.84, 1.08] with an R of 0.83. The 2022 comparison offers a slope of 0.95 [0.81, 1.08] with an R of 0.94. The results suggest a 30% mean percentage error and a detection limit of 200 kg of methane per hr at 12960 m cruise altitude above ground (13850 m above sea level). The results validate the emission estimates for methane point sources from oil and gas infrastructure using $XCH_4$ data from the MethaneAIR imaging spectrometer, based on blinded-volume controlled release experiments and comparing two algorithmic approaches for images of uncontrolled sources. Our study supports the application of our emission estimation algorithms to the upcoming MethaneSAT satellite after adjusting for the lower spatial resolution and higher signal-to-noise ratio, given appropriate testing and validation of the satellite. This study is limited by the number of collected data (21 data points on two days, 43 points on four days) from two locales; further evaluations in a variety of conditions are needed to establish detection limits and accuracy more fully.

This study tackles the challenge of quantifying the uncertainty of emissions due to the effects of eddy-scale variability on remote sensing observations. Our methods explicitly estimate the influence of eddies on the values derived for each emission source of interest. The *mIME* captures the variation across simulated cycles. The DI averages the flux information from the $XCH_4$ image across spatial scales that capture plume size and shape modulation by eddy scale motions. From our limited data, our *mIME* and the original application of IME by Varon et al. (2018) perform equally well, provided that local wind information is available. Our *mIME* performs better than an IME using only mesoscale-scale winds from HRRR. Thus, the *mIME* provides an alternative to local wind data, which is typically not available, and it also produces error estimates associated with the wind variability near the source of interest.

We conclude that the new MethaneAIR imaging spectrometer supports accurate emission rate estimates for high emitting methane sources above $200 - 500$ kg/hr detection limit when using the Integrated Mass Enhancement algorithm of Varon et al. (2018) modified for our system (*mIME*) or when applying our spatially oversampled application of the divergence integral. The sensor and analysis methods described here provide the greenhouse gas emission quantification community with improved approaches to consider the variability from the eddies. They establish the foundation for applying MethaneAIR and future MethaneSAT data to identify and quantify methane emissions to the atmosphere.

*Code and data availability.* The datasets and code generated and analyzed during the current study are available at https://github.com/ju21u/mair_controlled_release.

*Author contributions.* Author contributions. AC led this project. AC generated the LES, performed the mIME & ratio method, and wrote the manuscript with comments and revisions from all authors. SW supervised AC. MS led the DI analysis. TL helped AC develop the WRF-LES with additional LES insights from YL. SCW, MS, JSB, JEF, XL, KC, YL, RG, and MO helped to formulate the analyses. SCW, CCM, EC,

KC, AHS, JSW, BL, and SR prepared the MethaneAIR data used as inputs of quantification algorithms. RG and MO characterized facility-related bottom-up emission estimates. JSR, EDS, and AB were responsible for blinded-volume controlled release experiments on the ground. MS was responsible for the DI estimates. AC was responsible for the mIME estimates. AC prepared all figures for the manuscript. Everyone reviewed and edited the manuscript. Besides the first, second, and last, the authors were listed randomly and grouped by affiliations.

*Competing interests.* JSB, MO, and RG are employees of the Environmental Defense Fund (EDF). MethaneSAT LLC is a wholly-owned subsidiary of EDF. AC, MS, JEF, CCM, JSW, SR, and SWC received research funding from EDF

*Acknowledgements.* This work was supported by NSF EAGER grant 1856426 to Harvard University and by funding from the Environmental Defense Fund and Harvard University. The controlled methane release experiment was funded by ExxonMobil, the Stanford Strategic Energy Alliance, and the Stanford Natural Gas Initiative, an industry consortium that supports independent research at Stanford University. This work benefitted from discussions with S. Aminfard. We gratefully acknowledge the contributions of the flight crew and staff of NSF's Research Aviation Facility and discussions with the MethaneSAT Science Advisory Group and with S. Hamburg, Chief Scientist of the Environmental Defense Fund.

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
