# Peer review of "Methane Point Source Quantification Using MethaneAIR: A New Airborne Imaging Spectrometer"

_EGUsphere, 2023_

## Author Response (AR1)

**Author's Response: Post-Discussion Review**

August 23, 2023

**1   R1 General Comments**

This valuable case study analyses data from the airborne imaging spectrometer MethaneAIR, the aircraft precursor instrument for MethaneSAT. In addition to a comprehensive explanation of the methane retrieval algorithms, backed up by a detailed supplement, it contains two interesting and important highlights: overflights of a controlled methane release experiment to strengthen the confidence in the method and to quantify its detection limit, and complementary atmospheric modelling to handle the issue of methane plumes being distorted by atmospheric turbulence. The paper is well written and presents helpful figures and tables. I have the following minor comments.

**1.0.1   Author's response**

Thank you.

**1.0.2   Author's changes**

Please see the updated version of the manuscript.

**2   R1 Specific Comments**

**2.1   Introduction**

**2.1.1   R1's Comment**

It is good to hear about the positive development towards MethaneSAT, yet we should not forget that imaging spectroscopy has limitations. The intro would benefit from short explanations on how MethaneAIR/SAT will cope with variable or low surface albedo, the presence of aerosol in some of the plumes, the necessity to use $CO_2$ as a retrieval proxy, and the necessary instrument design tradeoff between high spectral and high spatial measurement resolution. Also, it is OK to cite many similar remote sensing techniques, yet IPDA lidar is missing.

**2.1.2   Author's response**

We can add more citations.

**2.1.3   Author's changes**

Line 37 - 39

**2.2   Section 2.1**

**2.2.1   R1's Comment**

You could/should mention that tropopause height variations influence the methane column due to less methane in the stratosphere. This may play a role at HIAPER flight altitudes and will certainly affect Methane-SAT data.

**2.2.2   Author's response**

We agree.

**2.2.3 Author's changes**

Line 119

**2.3 Scales**

**2.3.1 R1's Comment**

Figures 1 and 2 would benefit from displaying a km scale and an arrow indicating the approximate wind direction.

**2.3.2 Author's response**

We should.

**2.3.3 Author's changes**

Updated figures.

**2.4 Line 140**

**2.4.1 R1's Comment**

1.5 standard deviations of the inflow values, above the median value of the inflow. . .

**2.4.2 Author's response**

We can rewrite the sentence that way.

**2.4.3 Author's changes**

Line 165 - 166

**2.5 Line 141**

**2.5.1 R1's Comment**

Setting values below that value to NA.

**2.5.2 Author's response**

We can rewrite the sentence that way.

**2.5.3 Author's changes**

Line 167

**2.6 Equations 1 and 2**

**2.6.1 R1's Comment**

for the sake of logic I would place Eq 2 before Eq 1.

**2.6.2 Author's response**

We can do that.

**2.6.3 Author's changes**

Switched the order between Eq 1 and Eq 2 in the main text.

**2.7 Lines 158-160**

**2.7.1 R1's Comment**

They are a repetition of lines 150-152; should be removed.

**2.7.2 Author's response**

We agree.

**2.7.3 Author's changes**

Removed the repeated sentence.

**2.8 Line 219**

**2.8.1 R1's Comment**

"... a replacement for each synthetic observation." I do not understand this. Please explain more precisely with what you replace the original inflow pixels.

**2.8.2 Author's response**

We can rewrite the sentence for better clarity.

**2.8.3 Author's changes**

Line 252 - 255.

**2.9 Section 2.6.1**

**2.9.1 R1's Comment**

At which altitude (agl) was the methane released? The altitude plays a role in subsequent (turbulent) mixing and plume development. The LES release altitude should be selected accordingly.

**2.9.2 Author's response**

We can include the suggested information.

**2.9.3 Author's changes**

Line 291 - 292.

**2.10 Line 324**

**2.10.1 R1's Comment**

Typical emission rates were ...

**2.10.2 Author's response**

We can fix this.

**2.10.3 Author's changes**

Line 377

**2.11 Figure 4**

**2.11.1 R1's Comment**

There is an inconsistency with Line 341 where you state that the correlation statistics are computed for emissions exceeding 500 kg/hr. The figure also shows emissions ¡ 500 kg/hr. You should at least display those differently, or remove them from the figure. Also, I do not understand the red square: why does it represent "no detection" since it lies at about 700 kg/hr and therewith well above any detection limit?

**2.11.2 Author's response**

We made a mistake when we made that plot. The red square should not be there. We wanted to show emissions from the controlled release too. Estimates under 500 kg/hr can be open circles. Those above 500 kg/hr can be solid circles. We can also update the figure description to make it clearer.

**2.11.3 Author's changes**

Updated the figure and the caption.

**2.12 Line 395**

**2.12.1 R1's Comment**

The reference for Chan Miller et al is missing.

**2.12.2 Author's response**

Agree. Chan Miller et al. will be submitted by the end of this month.

**2.12.3 Author's changes**

Leave Chan Miller et al. 2023 there. Hopefully, we can update the reference as soon as the paper is submitted to AMT.

**2.13 Supplement, S1.1**

**2.13.1 R1's Comment**

I am afraid that the coefficients a and b depend on the ambient weather, or more precisely, on the turbulence conditions. Then, the relationship between $U_{eff}$ and $U_{10m}$ would lose its universality?

**2.13.2 Author's response**

We agree. We can include this concern in S1.1.

**2.13.3 Author's changes**

Updated S1.1

**2.14 Supplement, S1.1**

**2.14.1 R1's Comment**

S1.1: what do you mean by overfitting? This term is not familiar to me.

**2.14.2 Author's response**

We can elaborate what we meant by overfitting.

**2.14.3 Author's changes**

Updated S1.1

**2.15 S1.2**

**2.15.1 R1's Comment**

before equation S7: Combining equations S5 and S6... (not A5 and A6)

**2.15.2 Author's response**

We can combine the two equations.

**2.15.3 Author's changes**

Combined.

**2.16  S1.3**

**2.16.1  R1's Comment**

I am afraid that the ratio method is only applicable under the condition that the wind velocity is correct?

**2.16.2  Author's response**

Correct.

**2.16.3  Author's changes**

Added the limitations to the end of S1.3

**2.17  Figures S1 and S2:**

**2.17.1  R1's Comment**

Add the Stanford release rates for reference. I would plot lines with symbols instead of bars. Maybe merge both figures for better comparison? What do you mean with "endpoints"?

**2.17.2  Author's response**

We want to keep the flag information in the plots. The Stanford data are already shown in the table.

**2.17.3  Author's changes**

Updated the description of endpoints in S3.4 session.

**2.18  Figures S5 and S6:**

**2.18.1  R1's Comment**

Maybe merge both figures for better comparison, plotting lines instead of bars?

**2.18.2  Author's response**

We want to keep the flag information in the bar plots. It would be harder to do that with line plots.

**2.18.3  Author's changes**

No change.

**3  R2 Conceptual/Larger concerns**

**3.1  Missing information/reference on the retrieval**

**3.1.1  R2's Comment**

This paper is focused on the evaluation of quantification of point source emissions, which is a reasonable and appropriate scope. However, we are missing critical information on the retrieval product used to evaluate some elements of the manuscript. Chan Miller 2022 is cited repeatedly as a reference for the retrieval used, however there is not a journal or DOI listed with this reference. Extensive google scholar searching shows no such article. I do find two useful/relevant articles not cited, Conway et al., AMTD 2023 and Staebell et al., AMT 2021 that show the data processing and spectral calibration of MethaneAir respectively (those both should be cited here).

This leaves a rather problematic gap, as we have no information on the retrieval or how the XCH4 values are produced. Are these true total column values or just below the aircraft partial columns? What is the averaging kernel? This latter question is critically important as the quantification methods all do no discuss averaging kernel/sensitivity and implicitly assume that MethaneAir has a uniform averaging kernel, at least in the boundary layer. Is this true? Or should some correction be applied in the quantification to account for different sensitivity in the boundary layer (like in Wunch et al., GRL 2009)? I have further question on stability of retrieval and sensitivity to sampling – are all the flight legs getting the same background value and not have any aliasing that can manifest when combining the flight lines.

**3.1.2 Author's response**

Sorry about the confusion. The Chan Miller 2023 will be submitted to AMT by the end of this month (URL pending). I can share the current draft with R2. I agree that we missed the references to Conway et al., AMTD 2023 and Staebell et al., AMT 2021. We shall include them in the updated manuscript.

Regarding the retrieval algorithms, we use true columns comprising the paths from space to the ground and the ground to the aircraft ( 12 km). However, since most of the excess methane is within the boundary layer (below the aircraft), we assume that the averaging kernel is applicable for the lowest kilometers. We will include a graph showing the averaging kernel, which is slightly larger near the surface than in the upper atmosphere. The detailed discussion should be in the Chan-Miller et al. We didn't have a uniform averaging kernel. We use multilayer retrievals that include the shape of the averaging kernels.

**3.1.3 Author's changes**

Changed the reference to Chan Miller and mentioned Conway et al., AMTD 2023 and Staebell et al., AMT 2021. We also added more information regarding the averaging kernels (Line 117 - 129).

**3.2 Slope fitting**

**3.2.1 R2's Comment**

I appreciate there is some nuanced discussion of York fitting versus OLS. For comparison with the controlled release, this is a scenario where OLS would strike me at first as most appropriate – the x-axis should be very well known and the y-axis have relatively large uncertainty. I'm surprised in Figure 3 that the error bars appear comparable on both axes – but I don't know what those error bars represent and if they are the same. I'd like to see a little more concluding discussion on the slope methods. Right now it reads a bit as though the authors internally disagree on what slope method is correct, and presented both with a lean towards the York. But what should the community draw from this? To me, OLS would still be the correct approach for controlled releases provided the error is ¡ 3 times that of the tested airborne/satellite method. In this example perhaps there is a special case where the metered release had higher error?

**3.2.2 Author's response**

Two main points that we want to convey. 1. The community should be aware of the errors from the releases. In some cases (such as this controlled release experiment), errors are comparable. Thus OLS is inappropriate. We plan to add a second set of releases in 2022. The Stanford team had improved the instrumentation. In this new set of releases, their instrument errors are much smaller than our estimates. 2. York Fit is more appropriate as the regression was designed to have errors for both the x-axis and y-axis variables. In the second case where the Stanford errors are close to being negligible, the YorkFit is still appropriate and gives similar answers to OLS.

**3.2.3 Author's changes**

No changes.

**3.3 False positive/false negative rate?**

**3.3.1 R2's Comment**

Can anything be said about false positive or false negative rate from the data collected?

**3.3.2 Author's response**

We can add the rates.

**3.3.3 Author's changes**

We added the rates to S2.1.

**3.4 Care needed with extrapolation to MethaneSAT**

**3.4.1 R2's Comment**

At many times, particularly in the intro, there is an emphasis on the similarity and extensibility of this work to MethaneSAT. This needs to be presented with precision, as the work here demonstrates that these point source quantification approaches can work with appropriate imaging spectrometer data. So fair to say based on design specifications, this is extensible to MethaneSat, but we do not yet know what MethaneSat spectra will look like. (lines 41-43 for example). Also, it isn't indicated in this paper, but I had the impression different detectors are used in MethaneSAT and MethaneAIR, which can lead to significant differences. If this is so, it should be acknowledged and the similarities between the two instruments should not be overstated.

**3.4.2 Author's response**

We agree. We can acknowledge the differences and challenges.

**3.4.3 Author's changes**

Line 24 - 25, Line 46.

**3.5 Care needed with discussion of 'independent' winds**

**3.5.1 R2's Comment**

Care needed with discussion of 'independent' winds, and extensibility of results/approach outside US where HRRR is not available: In the abstract (line 17) it is stated the independent winds are used with different approaches. In fact, HRRR is foundational in both the LES and the DI approaches. I recognize driving the LES with HRRR is not ideidentical to using HRRR directly, but in the end the winds are based on the same input drivers. This should be clearly acknowledged. Further, the implications of this should be discussed as this is not an independent wind comparison. Also, HRRR is not available outside the US, so it should be discussed the impact that would/could happen for both LES and DI approaches with different/coarse resolution/less accurate wind fields. The impacts will be different for the two approaches. And the LES model will also be dependent on inputs for surface topography and energy balance which might be worse outside the US. There is a broader question embedded here in the scalability and extensibility of the LES approach.

**3.5.2 Author's response**

Agree that we should acknowledge the fact that DI and mIME used similar HRRR driver winds. There is a broader question embedded here in the scalability and extensibility of the LES approach. We will also acknowledge that the LES might not be able to scale globally.

**3.5.3 Author's changes**

Line 17. Line 148 - 151.

**4 R2 Specific comments/line by line**

**4.1 abstract line 17**

**4.1.1 R2's Comment**

States that LES and DI use different wind products – please correct this statement.

**4.1.2 Author's response**

Will do.

**4.1.3 Author's changes**

Line 17.

**4.2 abstract concluding sentence**

**4.2.1 R2's Comment**

This work doesn't demonstrate potential of "our instruments". It demonstrates potential of methaneAIR and suggests that the quantification method should be transferrable to the satellite if the satellite meets design specs.

**4.2.2 Author's response**

Will fix this.

**4.2.3 Author's changes**

Line 24 - 25.

**4.3 Around lines 35**

**4.3.1 R2's Comment**

Sentinel 2 , WorldView 3 were not designed for methane sensing but do it anyways. Also are missing PRISMA here. It is important to clarify whether the instrument was designed for methane use case or not. Actually aviris-ng was also not designed for methane either (though carbonmapper will be). One point being that sensors designed/dedicated to this purpose should outperform sensors reanalyzed for this.

**4.3.2 Author's response**

We can include PRISMA here. We will also make sure to mention whether the instrument was designed for methane use.

**4.3.3 Author's changes**

Line 36 - 44.

**4.4 Line 41**

**4.4.1 R2's Comment**

: "nearly identical spectroscopy". Don't the two instruments use different detectors? Don't overstate the similarities.

**4.4.2 Author's response**

We will replace the word "nearly" with "very similar."

**4.4.3 Author's changes**

Line 46.

**4.5 Line 43**

**4.5.1 R2's Comment**

Methanesat is designed to have the capabilities — but we do not yet know it will (fingers crossed).

**4.5.2 Author's response**

We include this change.

**4.5.3 Author's changes**

Line 49.

**4.6 Line 47**

**4.6.1 R2's Comment**

Clarify, you are validating emissions estimates here, you are not validating concentrations in this paper.

**4.6.2 Author's response**

We will fix this.

**4.6.3 Author's changes**

Line 53 - 54.

**4.7 Line 65**

**4.7.1 R2's Comment**

line 65 typo "dein"

**4.7.2 Author's response**

We will fix this.

**4.7.3 Author's changes**

Line 71.

**4.8 Line 74**

**4.8.1 R2's Comment**

Using HRRR is restricted to US. If wanting to do this elsewhere, what happens? Some discussion of input met field dependence/importance?

**4.8.2 Author's response**

We tried estimating the emissions using different wind products. It's hard to generalize which product is better or worse than the others.

**4.8.3 Author's changes**

Line 148 - 151.

**4.9 Line 88-89**

**4.9.1 R2's Comment**

What is interference from nearby activity? Wouldn't one say distinguishing amongst other activity important?

**4.9.2 Author's response**

In this case, interference from nearby sources means methane enhancement from nearby sources that can't be separated from the source of interest.

**4.9.3 Author's changes**

Line 90 - 91.

**4.10 Table 1**

**4.10.1 R2's Comment**

Why start on RF04 - what happened with RF01-03? What was successful flight fraction/instrument duty cycle?

**4.10.2 Author's response**

RF04 is simply an identification of the research flight of the MethaneAIR campaign series. The first three research flights (RF01 - RF03 ) were successful engineering flights. Our main purposes weren't for methane data collection, but rather the making sure that our instrument was working properly. We didn't report data from the engineering flights.

**4.10.3 Author's changes**

Line 96 - 99.

**4.11 Figure 1**

**4.11.1 R2's Comment**

Is this true total column XCH4 or partial column? What are gaps in Figure 1b? Maybe show faint flight lines so we can see the flight pattern.

**4.11.2 Author's response**

Those gaps are from filtered shadows, clouds, and water basins. We will include the flight tracks in the supplement.

**4.11.3 Author's changes**

Figure 1 caption and S5 in the supplement.

**4.12 lines 105-110**

**4.12.1 R2's Comment**

what is the impact of the gridding/smoothing choice on methane quantification? The gaussian filter is somewhat arbitrary and understanding the impacts of the smoothing on emissions estimates is important. What happens with no filter? What happens with filters that are more/less aggressive with different correlation lengths?

**4.12.2 Author's response**

We have tried various smoothing/gridding options. They don't really matter much. If we don't do smoothing, we would not account for the spatial oversampling. We would get higher noise level and harder to detect small emissions.

We chose the Gaussian filter because it conserves mass and is a standard way to denoise an image.

**4.12.3 Author's changes**

Line 132. Please also refer to Chan Miller et al. 2023 for more information.

**4.13 Line 112-113**

**4.13.1 R2's Comment**

What are the implications of assuming source location was identified? This may be big real world challenge and more discussion in warranted.

**4.13.2 Author's response**

By assuming that source locations were identified, we have a lower chance of having false positives. Identifying source locations is challenging for various reasons. In addition, our algorithm for plume detection is still developing. We want to be explicit that our plume detection work is not fully automated yet.

**4.13.3 Author's changes**

Line 138 - 139.

**4.14  2.3.1**

**4.14.1  R2's Comment**

Does the entire LES approach require knowledge of source site to run in its forward mode? How does this impact the feasibility to run for lots of unknown source locations and does this require first an algorithm to identify source location? Building off this and computational requirements, is this LES approach scalable?

**4.14.2  Author's response**

We have a plume finder algorithm based on the DI approach to identify potential sources. We plan to run LES for only important targets identified by the plume finder algorithm. You need to specify the domain of the LES and make sure that the source is within that domain (10 km x 10 km at the moment, could be larger). With the current setup, it only takes less than 4 hours to complete the run. The computational cost is much cheaper compared to the L2 data processing cost. Regarding scalability, once we automatically cross-check the DI-identified sources and the infrastructure inventories, we should be able to include the mIMe method in the pipeline.

**4.14.3  Author's changes**

No changes in the paper.

**4.15  Line 147**

**4.15.1  R2's Comment**

Is the LES resolution coarser b/c of computation? Why even in this limited test/example cases could it not be run at comparable scale to the observations?

**4.15.2  Author's response**

Correct, the compute time would be at least 25 times longer than the current setup. We did try running LES at   12 m by 12 m resolution. The plumes look similar to the 111.111 m by 111.111 m resolution. So, we chose to use 111.11 m by 111.11 m to be consistent with the resolution that we plan to use operationally. Note, this version of LES uses a closure approach in the boundary layer that renders it to be impossible to resolve plumes under 50 m scale.

**4.15.3  Author's changes**

No changes.

**4.16  Line 155**

**4.16.1  R2's Comment**

It is a bit unclear what you do for Ueff in the main text, is it the Varon equation or do you derive yourself from LES model? You state in the supplement you end up using Varon. Can you evaluate these assumptions/parameterization at all with the LES runs you have? What type of errors might be imposed? What is the averaging kernel? All the flux quantification assume uniform averaging kernel of 1.

**4.16.2  Author's response**

The results shown in the manuscript were based on Varon's equation. I did derive the relationship between the Ueff and U from the simulation though. Since I only simulate one plume at a time, a representative relationship between the Ueff and U cannot be computed from our data.

**4.16.3  Author's changes**

We added more information regarding the averaging kernels (Line 117 - 129).

**4.17  Line 175**

**4.17.1  R2's Comment**

How is the wind speed rotated?

**4.17.2 Author's response**

It depends on the source and time span. The winds can rapidly rotate under some conditions over a certain time span. The winds were rotated by (1) finding the second moment of inertia of the plume (2) rotating the HRRR wind direction to be coinciding with the x-axis.

**4.17.3 Author's changes**

Line 207 - 208.

**4.18 Figure 2**

**4.18.1 R2's Comment**

How are the different rectangles selected? Custom/by eye for each source? Why only step outward in direction of plume flow?

**4.18.2 Author's response**

We used the plume finder algorithm to identify the location of the plume. Then the upwind boundary was selected to be close to the upwind of the source. The box grows by 1 pixel in each step except for the upwind direction, where it moves over by 1/4 pixel. This is not done by hand.

**4.18.3 Author's changes**

Updated S1.2.

**4.19 Variability**

**4.19.1 R2's Comment**

Could some variability be addition for other sources/sites? It looks possible from the disconnected plume in 2b. Why does the flux increase after 700m, and why would that be the only place for other sources?

**4.19.2 Author's response**

Correct. The increase after 700 m is due to interference from CH4 emissions nearby or under the original plume.

**4.19.3 Author's changes**

Updated Figure 2's caption.

**4.20 section 2.4**

**4.20.1 R2's Comment**

A lot of 'we compared..' but then not display of comparison or discussion of how that comparison is used. Leaves me wondering a bit what was learned/demonstrated with these comparisons. In places diffuse sources capability is discussed but not really demonstrated with this paper.

**4.20.2 Author's response**

We can elaborate more.

**4.20.3 Author's changes**

Line 227 - 229.

**4.21 Line 264**

**4.21.1 R2's Comment**

Is methane air 10x10 or 20x20?

**4.21.2 Author's response**

The boundaries between native pixels are separated by 5 meters across track and by 25 meters along track. The point spread function is roughly two and a half pixels wide. The image oversamples spatially. We project the images onto a grid of 10 by 10 m across and take Gaussian filter into account for spatial oversampling. In an updated version of the retrievals we didn't use here, the spatial oversampling is accounted for in the gridding process, and we didn't use the Gaussian filter. Similar results are obtained.

**4.21.3 Author's changes**

Line 300.

**4.22 Line 265**

**4.22.1 R2's Comment**

This would suggest the authors should then state that with this sample approach other sources within 1km impact ability to quantify emissions... Please show the fully blinded 1:1 figure with slope/fitting as well as the after unblinding. It's not really fair to call the release range small b/c lower than 1,000 kg/hr — 1,000 kg/hr is a very large emissions rate.

**4.22.2 Author's response**

We didn't update the emissions after the unblinding. We only used the unblinding results to develop the decision tree to filter out potential low-quality emission estimates. The upper bounds for other teams that participated in the controlled release experiment were well above 1000 kg/hr (Sherwin et al. 2022).

**4.22.3 Author's changes**

No changes

**4.23 Fig: 3**

**4.23.1 R2's Comment**

How is the point at 250 flagged as below detection limit when the measured emission was over 200, higher than three other non-flagged points? Are the flagged points included in the fit? What are the reported error bars shown here? is it true that the meter error on the ground is comparable to the retrieval? in Sherwin et al it looks like much smaller meter uncertainty

**4.23.2 Author's response**

For the 2021 experiment, the errors were comparable. In Fig 3, unlike the no-detect (red square, only one release), the flagged data (red circles) are the small plumes that have a low signal-to-noise ratio decided by the decision tree. All points are included in the fit. The errors are 95% CI reported by the Stanford team and our estimates.

**4.23.3 Author's changes**

Updated the caption for Figure 3 and referred to S11.

**4.24 Line 332**

**4.24.1 R2's Comment**

While invoking rapture is entertaining, I believe you mean rupture.

**4.24.2 Author's response**

You are right.

**4.24.3 Author's changes**

Changed to rupture.

**4.25 Line 356**

**4.25.1 R2's Comment**

Need to add with appropriate testing and validation of the satellite...

**4.25.2 Author's response**

Will do.

**4.25.3 Author's changes**

Line 413.

**4.26 Line 366**

**4.26.1 R2's Comment**

Well, but it is computationally intensive and requires accurate input winds (HRRR) and accurate representation of surface topography and energy balance as well...

**4.26.2 Author's response**

Agree.

**4.26.3 Author's changes**

No changed.

**4.27 Eq S5**

**4.27.1 R2's Comment**

S5: so what happens with negative values? since you use the average to subtract you will have a lot of negative values... could you expand the discussion in the final two paragraphs?

**4.27.2 Author's response**

The negative values are screened out by using a clumping algorithm and eliminating clumps smaller than 20 pixels (stated S4).

**4.27.3 Author's changes**

More description was added to S1.2 in the supplement.

**4.28 OLS**

**4.28.1 R2's Comment**

Conceptually OLS would make the most sense.

**4.28.2 Author's response**

We already commented that the errors in the 2021 controlled releases are not negligible. Thus, York provides the maximum likelihood estimator.

**4.28.3 Author's changes**

No changes.

**4.29 heteroskedasticity**

**4.29.1 R2's Comment**

Unless metered errors were larger than anticipated and correlated with release rate the heteroskedasticity test is surprising.

**4.29.2 Author's response**

That was the case.

**4.29.3 Author's changes**

No changes.

**4.30 Further question on quantification related to the retrieval**

**4.30.1 R2's Comment**

What about topography changing the column air mass, and the air mass in the pbl. How does that come into play in these quantification measures and does that need to be addressed/discussed?

**4.30.2 Author's response**

The retrievals take account of the change in the topography since it's multilayer retrieval. For these comparisons, these are not significant topography near the point of the releases. For more information, please refer to Chan Miller et al. 2023 (soon to be submitted)

**4.30.3 Author's changes**

We added more information regarding the averaging kernels (Line 117 - 129).

---

## Author Response (AR2)

**Author's Response: Associate Editor Decision**

September 29, 2023

**1 Introduction**

Introduction: it is good to hear about the positive development towards MethaneSAT, yet we should not forget that imaging spectroscopy has its limitations. The intro would benefit from short explanations on how MethaneAIR/SAT will cope with:

- variable or low surface albedo,

- the presence of aerosol in some of the plumes,

- the necessity to use $CO_2$ as a retrieval proxy, and

- the necessary instrument design tradeoff between high spectral and high spatial measurement resolution.

**1.0.1 Author's response**

While we believe that not all of these concerns should be incorporated into the introduction, we do acknowledge the existence of certain inadequacies in our explanations. We have addressed these concerns and integrated them into the paper as outlined below.

**1.0.2 Author's changes**

For the albedo, aerosol, and $CO_2$ proxy retrieval, please see lines 121 - 122 and 120 - 136. For the tradeoff, please see lines 42 - 45 in the introduction.

**2 Figures**

Figures 1 and 2 would benefit from displaying an arrow indicating the approximate wind direction.

**2.0.1 Author's response**

We can add that.

**2.0.2 Author's changes**

Please see the updated Figures 1 and 2.

**3 MethaneAIR vs MethaneSAT Discussion**

Furthermore, I agree with referee 2 that the discussion of differences between MethaneSat and MethaneAir is too short. You only changed "nearly identical spectroscopy" to "very similar spectroscopy", which is a too qualitative statement. It is important in the context of this manuscript to understand how well the results are transferable to MethaneSat - or not.

**3.0.1 Author's response**

We can comment more on the transferability and differences.

**3.0.2 Author's changes**

Please see lines 53 - 55 in the introduction and the comparison table in the supplement.